# Is SARS-CoV-2 Vaccination of Subjects with a Prior History of Allergies Dangerous? Experiences in the Veneto Region of Italy

**DOI:** 10.3390/vaccines11030574

**Published:** 2023-03-02

**Authors:** Silvia Cocchio, Gloria Girolametto, Alice Pierobon, Patrizia Furlan, Emanuela Destefani, Lorenzo Bulegato, Antonio Stano, Silvia Fietta, Annachiara Poletto, Vincenzo Baldo

**Affiliations:** 1Department of Cardiac Thoracic Vascular Sciences and Public Health, University of Padua, Via Loredan 18, 35100 Padova, Italy; 2Department of Prevention of Local Health Unit n. 7, 36061 Bassano del Grappa, Italy

**Keywords:** SARS-CoV-2, vaccination, allergies

## Abstract

Adverse events after SARS-CoV-2 vaccinations have caused alarm to some individuals with previously diagnosed allergies. The aim of this study was to investigate whether the risk of adverse reactions was actually higher in this subgroup. To this end, we carried out an observational descriptive analysis of vaccines administered in a “protected setting” in the Veneto region of Italy between December 2020 and December 2022. Reactions were classified using systemic organic classification (SOC), and their severity was assessed using the criteria of the Italian Drug Agency (AIFA). A total of 421 subjects were vaccinated with 1050 doses, 95.0% of which were administered without adverse events. In all, 53 subjects reported 87 SOC reactions (1.6 reactions/person), and 18.3% of these reactions were severe. One person was hospitalized, but all subjects enjoyed complete remission. Reporting rates were 9.0%, 3.1%, and 1.2% for first, second, and third doses, respectively. The most frequent reactions involved the respiratory system (2.3%), the cutaneous and subcutaneous systems (2.1%), and the nervous system (1.7%). Multivariate analyses (adjOR (95% CI)) revealed that the probability of experiencing at least one reaction significantly declined with increases in age [0.95 (0.94–0.97)] and in the number of doses received, i.e., 75% [0.25 (0.13–0.49)] for second doses and 88% [0.12 (0.04–0.39)] for third doses. These results indicated that vaccinations could be safely administered; few reactions were reported, and there were no permanent adverse outcomes.

## 1. Introduction

A linear sequence involving several steps over a period of many years is usually necessary for the development of a new vaccine [1]. The sudden outbreak of the SARS-CoV-2 pandemic prompted a rapid scientific response and led to an unprecedented effort to produce a vaccine against this virus in a very short time. This took place in a context of limited previous clinical experience with any coronavirus vaccine. Multiple vaccine development phases were carried out in parallel, and new vaccine technologies were used, some of which had never previously been used for licensed vaccines [2]. Both of the initial COVID-19 vaccines authorized by the Food and Drug Administration (FDA) and the European Medicines Agency (EMA) between December 2020 and January 2021—the BNT162b2 and the mRNA-1273—have been attested as safe in clinical trials [3]. These vaccines consist of a nanoparticle-encapsulated lipid that encodes the full-length SARS-CoV-2 spike (S) glycoprotein, and so they are characterized by the technology of nucleoside-modified messenger RNA (mRNA) [1]. However, even as mass vaccination campaigns were put into full operation worldwide, vaccine-related reactions were also increasingly reported, despite the benign course of the vaccination program in general [4]. In terms of allergic reactions, the relevant unexpected anaphylaxis rate was reported to be 4.7 cases per 1 million administered doses of BNT162b2 vaccine [5], and 2.5 cases per 1 million doses of mRNA-1273 [6]. It was reported that 90% of anaphylactic reactions reported after BNT162b2 administrations were among females, and 81% were in individuals with previously diagnosed allergic conditions [7]. It was also reported that the relative incidence of anaphylaxis following mRNA COVID-19 vaccine administrations was between two and seven times higher in subjects with a prior history of allergies, compared with those with no such history [8]. Considering the vaccine program in Italy alone, the 11th AIFA COVID-19 vaccine report, released at the end of March 2022, reported 3 cases of anaphylaxis for every 1,000,000 doses of BNT162b2 vaccine administered [9], and 99 suspected post-vaccination adverse reactions for every 100,000 doses administered, regardless of the type of vaccine or the number of doses [9]. The trends and related rates identified in these reports were described as basically stable over time [9]. Most of the suspected adverse events were classified as non-serious (82.1%), though 17.8% were serious, and in 0.01% of cases, it was not possible to indicate the degree of severity. However, in most cases, the outcome involved either improvement or complete resolution of signs and symptoms [9].

Taking all these considerations into account, it is important to remember that in the clinical trials of both the BNT162b2 and mRNA-1273 COVID-19 vaccines, participants with a history of allergic reaction to any component of the vaccine or any allergy to other vaccines were excluded [7]. In the trials of BNT162b2, individuals with other allergies were also excluded from participation [7]. Within the context of a vaccine administration campaign during a full-scale global emergency, rising levels of concern among patients and physicians about allergic reactions to COVID-19 vaccines led the Center for Disease Control and Prevention (CDC) and other authorities to issue a number of recommendations concerning the safety of the new mRNA vaccines in patients with a history of allergic reactions [7]. Official European (and Italian) guidance for mRNA vaccine administration was issued stating that “people who already know they have an allergy to one of the components of the vaccine listed in […] the package leaflet should not receive the vaccine. People who have a severe allergic reaction when they are given a dose of mRNA vaccine should not receive subsequent doses” [9,10]. Although any allergens within the vaccines are yet to be determined definitively, it has been suggested that the polyethylene glycol (PEG) used for the construction of the nanoparticle-encapsulated lipid may be a likely candidate [4].

Despite the diffusion of all these recommendations, many individuals found that their vaccination was deferred by their physicians or by immunization teams because of “allergic concerns”. Uncertainty continues to surround this issue, particularly in regard to those with a history of anaphylaxis and/or multiple allergies. Moreover, reports of adverse events have prompted some individuals to publicly express concerns about the vaccination program, or to delay getting vaccinated themselves, or to oppose the administration of vaccines altogether [11].

There is no doubt that the worldwide COVID-19 pandemic has led to an unprecedented level of interest in vaccines, public health and related scientific research. Attention has been newly focused on the development of vaccines and on related regulatory review and safety monitoring issues; many public comments have been made on such matters, and not only by experts, using forms of communication previously rarely used in scientific fields, such as mass and social media [10]. In such a context, the clear, effective, and consistent communication of evidence and/or uncertainties is therefore essential to help and correctly advise people to approach the question of vaccination critically [12]. With this in mind, it is notable that very few studies to date have examined the safety and tolerability of COVID-19 mRNA vaccines in people with previously diagnosed allergies, or considered the severity of post-vaccination adverse effects among such individuals [13]. In this study, we analyzed COVID-19 vaccination adverse effects among individuals with a previous history of severe and/or multiple allergic reactions in a “protected setting” in the Veneto region of Italy, to determine if these subjects did or did not exhibit a higher risk of experiencing post-vaccination adverse effects than the rest of the population.

## 2. Materials and Methods

### 2.1. Study Design and Setting

An observational study was carried out to analyze reactions to COVID-19 mRNA vaccines administered between 27 December 2020 and 31 December 2022 in a “protected setting” in the city of Bassano del Grappa, in the Veneto region of Italy. The “protected setting” was located in the COVID-19 vaccination center which was established in front of the local Emergency Department of the city’s Public Hospital. Vaccine administration personnel were supported by an intensive/urgent care team who were present in an observational area and ready to intervene in case of adverse reactions. Emergency medications and instruments were available for use on an outpatient basis. Patients could approach the center on their own behalf or via a request by their general practitioner (GP).

### 2.2. Study Population

Only patients with a prior history of allergies were included in the study.

Patients were considered to be at high risk of allergic reactions if any one of the following criteria was exhibited: prior anaphylactic reaction to any drug or vaccine; multiple drug allergies; multiple allergies; or mast cell disorders.

All subjects included in the study were individuals who had not been vaccinated in a regular setting by their general practitioner (GP) or by a member of an immunization team because of concerns about allergic reactions. All were re-evaluated by means of a detailed tele-evaluation carried out by doctors in the Prevention Department who had expertise in vaccines.

The tele-evaluation included specific questions about prior allergic or anaphylactic reactions to one or more oral or injectable drugs or vaccines. For all such reactions, the name of the drug and the reaction were both recorded, and consideration was taken of all previous medical documents available, focusing on treatments used (for example, antihistamines and/or glucocorticoids and/or adrenaline) and whether hospitalization had been required. The tele-evaluation also covered allergies to insect bites and particular foodstuffs, as well as inhalant allergies and any other allergic conditions identified by the patient, or by the previous evaluation of an allergist. Subjects were also asked about any therapies which they might be undergoing, and also about any comorbidities, with particular attention paid to immune comorbidities such as chronic urticaria, mast cell disorders, or asthma. In cases of bronchial asthma, the grade of control was determined by means of the Global Initiative for Asthma (GINA) score, part A [14].

### 2.3. Immunization and Adverse Reactions

In this study, the vaccine used for all administrations was BNT162b2. All patients were immunized lying down on a stretcher by means of an intravenous cannula needle which was positioned before the immunization and removed only at the end of an observational period which lasted at least 60 min. Premedication was not recommended prior to receiving the first dose of the vaccine unless patients were regularly treated with the drugs concerned (for example, antihistamines); in such cases, patients were instructed to continue with their medication. Adverse events were identified and reported by doctors if they appeared during the first 60 min after vaccination. Prior to administration of second and third doses, active calls were made to all patients so that any late-onset adverse reactions could be recorded.

All adverse reactions were classified using preferred term names and/or the important medical event terms list (MedDRA version 25.0) [15]. These were then subclassified using systemic and organic classification (SOC, MedDRA version 25.0) [15].

Reactions were considered to be severe or mild-to-moderate, by applying AIFA criteria for the evaluation of reactions [16].

Outcomes were classified as either “complete resolution”, “improvement”, “not yet cured”, or “resolution with sequelae” using pre-established criteria in line with European and national legislation [16].

### 2.4. Ethical Consideration

According to Italian national guidelines (Ministerial Decree 18/03/1998), anonymized data may be analyzed and used in aggregate form for scientific studies without further authorization. Therefore, no formal ethics committee approval was needed for the present study. However, this study was fully compliant with the requirements of the latest version of the Declaration of Helsinki. All patients gave their consent, and all data were anonymized before analysis. The data were treated with full respect for confidentiality, in accordance with Italian legislation. Before the database was made available to the authors, all sensitive data concerning the patients considered in the study were replaced with anonymous codes, making it impossible to identify the individuals concerned.

### 2.5. Data and Statistical Analysis

Demographic dates, all relevant general medical history, a complete evaluation of previous allergies, the type of COVID-19 vaccine administered, the date of administration, details of any reported post-vaccination adverse reactions, and all possible infections with COVID-19, either previously or subsequently, were entered into a Microsoft Office Excel database.

Data were analyzed using the chi-square test and Student’s *t*-test (for unpaired data). A multivariable logistic regression was performed to assess the probability of having a post-vaccination adverse reaction; from this, adjusted odds ratios (adjORs) and corresponding 95% confidence intervals (95% CIs) were calculated. The covariates included in the model were demographic and individual variables (sex, age, number of doses administered, number and type of previous allergies). A *p* value of less than 0.05 was considered statistically significant. Statistical analyses were performed using SPSS Statistics, version 28.0.

## 3. Results

During the study period, 421 individuals with multiple prior allergic conditions were vaccinated with at least one dose of COVID-19 vaccine. This represented 0.26% of a population of 158,618 people who received at least one dose of COVID-19 vaccine in the same period. Of the 421 patients included in the study, 304 (72.2%) were females. The average age of the sample group was 51.8 ± 18.6 years, and the average age of females was significantly higher than males (53.1 ± 17.3 years vs. 48.6 ± 21.1; *p* = 0.041).

In total, 1050 doses of COVID-19 vaccine were administered: 421 as first doses, 388 (92.2%) as second doses, and 241 (57.2%) as third doses. Figure 1 shows the number of subjects vaccinated with successive doses and the numbers of adverse reactions reported. Overall, 53 subjects (5.0%) reported at least one post-vaccination adverse reaction, with an average of 1.6 reactions per person. Specifically, after administration of first doses of vaccine, 38 subjects (9.0%) reported at least one adverse reaction. These individuals experienced a total of 64 reactions, corresponding to 1.7 reactions per person in this group. Eight of these thirty-eight subjects experienced an adverse reaction within an hour of administration; specifically, between 10 min and 50 min. Three individuals experienced a reaction within 48 h, and two within 8 days. In the remaining twenty-four cases, the reaction time was not specified. After second doses of vaccination, 12 subjects (3.1%) reported a total of 19 adverse reactions (4.9%), giving an average of 1.6 reactions per person in this group. Four of these individuals experienced an adverse event within an hour of administration; specifically, between 28 and 40 min. Another subject experienced a reaction after 7 days. In the remaining seven cases, the time was not specified. Finally, after third doses, three subjects (1.4%) reported a total of four reactions (1.8%), giving an average of 1.3 reactions per person in this group. One subject experienced an adverse event within an hour of administration, specifically, after 20 min. In the remaining two cases, the onset time was not specified.

Table 1 shows the numbers of different reactions reported, in terms of SOC classification and numbers of doses administered. Overall, 997 individual doses of vaccine were administered without reactions. A total of 53 administrations (5.1%) resulted in at least one adverse reaction, with 87 such reactions recorded in total. More specifically, in terms of SOC classification, twenty-eight subjects (2.7%) reported one reaction, seventeen subjects (1.7%) reported two reactions, seven subjects (0.7%) reported three reactions, and one subject (0.1%) reported four reactions.

Figure 2 illustrates the 87 post-vaccination reactions in terms of the dose received (first, second, or third (booster)) and typology (SOC classification). In brief, post-vaccination reactions occurred after 15.2%, 4.9%, and 1.8% of first, second, and booster doses, respectively. The most frequent adverse reactions involved the respiratory system (2.3%), the cutaneous and subcutaneous systems (2.1%), and the nervous system (1.7%).

According to AIFA severity evaluation criteria, of the 87 reactions reported, 16 (18.3%) were severe. Six subjects (37.5%) manifested asthmatic crises, five subjects (31.3%) manifested dyspnea at rest, two subjects (12.5%) manifested angioedema, one subject (6.3%) manifested hypertensive crisis, one subject (6.3%) had a syncope, and one subject (6.3%) manifested inferior arts’ akinesia. One individual required hospitalization; however, all subjects who experienced adverse reactions also enjoyed a full subsequent remission, with complete resolution of all signs and symptoms. The mean age of the subjects who experienced at least one reaction was 43.6 ± 14.9 years. Mean age was 41.8 ± 10.9 among those with a serious reaction, and 44.4 ± 16.4 among the others, *p* = 0.556.

In Table 2, the multivariate analysis showed that the likelihood of experiencing at least one reaction significantly declined as the age of the vaccinated subject increased (adjOR(95%CI): 0.95 (0.94–0.97)). Adverse reactions were also significantly less likely with second doses of vaccine (adjOR(95%CI): 0.25 (0.13–0.49)) and also with third doses (adjOR(95%CI): 0.12 (0.04–0.39)).

## 4. Discussion

To date, there have been few studies in the literature concerning administration of COVID-19 vaccines to people with multiple previous allergic reactions, as recalled in anamnesis. In this study, 0.26% of the subjects vaccinated in our department were thought to be at risk of post-vaccination adverse events because of their allergic conditions. We sought to investigate the safety and tolerability of mRNA COVID-19 vaccines in these patients.

Our findings suggest that, although the precise risk factors for allergic reactions to mRNA vaccines have yet to be determined, with a good pre-vaccination evaluation and administration protocol, the majority of the subjects with previous multiple allergies can be safely immunized. In our study, almost all post-vaccination adverse effects were treated successfully at the immunization site with no requirement for hospitalization, and numbers of severe adverse reactions were in proportion with reported data for the Italian national population [9,17]. All severe reactions reported were completely resolved in terms of signs and symptoms after basic care provided almost exclusively in an outpatient setting.

The mean age of our sample group was 51.8 years. This is 5 years older than the mean age reported for Italian individuals reporting post-vaccination adverse events or suspected such events (46.9 years) [9]. As previously reported in the literature, older individuals are more likely to have a history of allergic conditions, though the reason for this is not fully understood [13]. It may be because older people have been exposed to more allergy-causing foods and medications during their longer lifetimes [13]. With this in mind, it was interesting that our data analyses indicated a protective role of age, because the likelihood of manifesting an adverse vaccine reaction declined significantly as the age of vaccinated subjects increased. Other studies have confirmed this result [13,18] and this is an important research finding given that COVID-19 case-fatality and hospitalization rates have always been strikingly higher in older patients, especially those aged 65 years or more [19].

From our local experience, another interesting finding was that reported numbers of adverse events were considerably lower after later doses of vaccine. This result was in line with AIFA reports [9,17]; however, it might be considered surprising in a subgroup of allergic patients. Any sensitization after administration of a first dose should be expected to favor the onset of allergic reactions after a second dose. A possible explanation, in line with AIFA suggestions, may be that greater caution was taken in administrations of second doses to subjects who reported reactions after the first, by modifying the method of administration (in a protected setting, for example) or by suspending the vaccination process entirely [17]. In contrast, it is known that suspected allergic reactions after a vaccine administration are not necessarily mediated by the immune system, at least not entirely. For example, there are several mimics of anaphylaxis described in the literature, including vasovagal syncope and vocal cord dysfunction [20]. Nevertheless, a temporal association between a vaccination and a subsequent event is a necessary, but not sufficient, criterion to establish that the vaccination was the cause of the event [17].

Despite the protected administration setting, and the evaluations of specialists, a considerable number of decisions to suspend the vaccine program were made either by doctors or autonomously by individuals themselves (17.2%). In fact, only one patient was expressly exempted by doctors from administration of later vaccine doses. This may suggests a high degree of vaccine hesitancy, in spite of the protected setting for vaccine administration. That said, it is evident that patients who have experienced previous allergic reactions and who then experience post-vaccination adverse events, especially in a normal context of vaccination without active calls, may unnecessarily avoid COVID-19 vaccine administration, raising the possibility of reduced levels of vaccination without any good scientific cause [20].

Considering possible limitations of this study, we should state that mechanisms of reactions such as tryptase levels or sensitivity to vaccine ingredients (as measured by the PEG 2000 cutaneous test) were not measured for all people reporting reactions, as these tests were not available in our medical center. In this area, our study needs to be supported by further investigations. In addition, we could not differentiate reported post-vaccination reactions in terms of onset times, even though it is known that allergic reactions to vaccines are either acute or late in onset [7]. All the acute adverse reactions were detected by doctors during the first 60 min after vaccination and so these were reported with certainty; however, we could not obtain similar assurance for late-onset reactions because most patients reported reactions by themselves, without a written medical evaluation. This meant that some reactions may not have been reported at all, leading to possible under- or over-estimation in the final study results. That said, it is important to remember that almost half of all allergic reactions reported in the literature are described as occurring within 15 min of vaccination (immediate hypersensitivity reactions) [21], and that the median interval from COVID-19 vaccine administration to the onset of reaction has been reported as 13 min [22]. We also note that the onset of symptoms in the most severe allergic cases is most likely to occur within 30 min of vaccination [22].

Finally, we would like to confirm that the initial evaluation was successfully carried out by our specialists by means of a tele-health visit, as suggested by other authors [12].

## 5. Conclusions

We enabled COVID-19 vaccination of patients with multiple previous allergies reported in anamnesis using a simple exemplar procedure which included a pre-vaccination optimal risk assessment anamnesis, the option of contacting a referral center or an allergist, and a safe environment for immunization with the facility to treat all types of reactions during a prolonged period of post-vaccination observation. This level of provision should be implemented in all medical settings to allow immunization for all, so that allergic patients enjoy access to the same publicly recommended vaccines as nonallergic patients, except when the risks associated with vaccination outweigh the gains. As this vaccine prevents a deadly disease and is the principal tool to control the pandemic, immunization of all the population, including those with an allergic history, if possible, is an important goal; overcoming or, at least, clarifying safety concerns, especially in regard to allergic conditions, is absolutely required to achieve this goal. Our data provide a real-life description of safe tolerability profiles of mRNA COVID-19 vaccines in patients with previous allergies; however, our findings need to be confirmed in larger studies.

## Figures and Tables

**Figure 1 vaccines-11-00574-f001:**
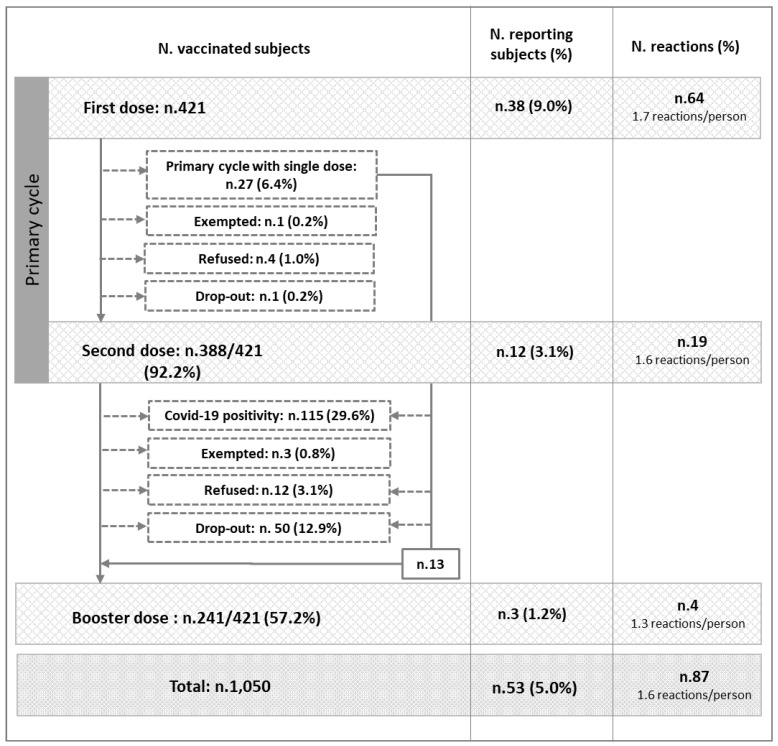
Distributions of subjects vaccinated, numbers of subjects reporting at least one post-vaccination adverse reaction, and numbers of post-vaccination adverse events reported.

**Figure 2 vaccines-11-00574-f002:**
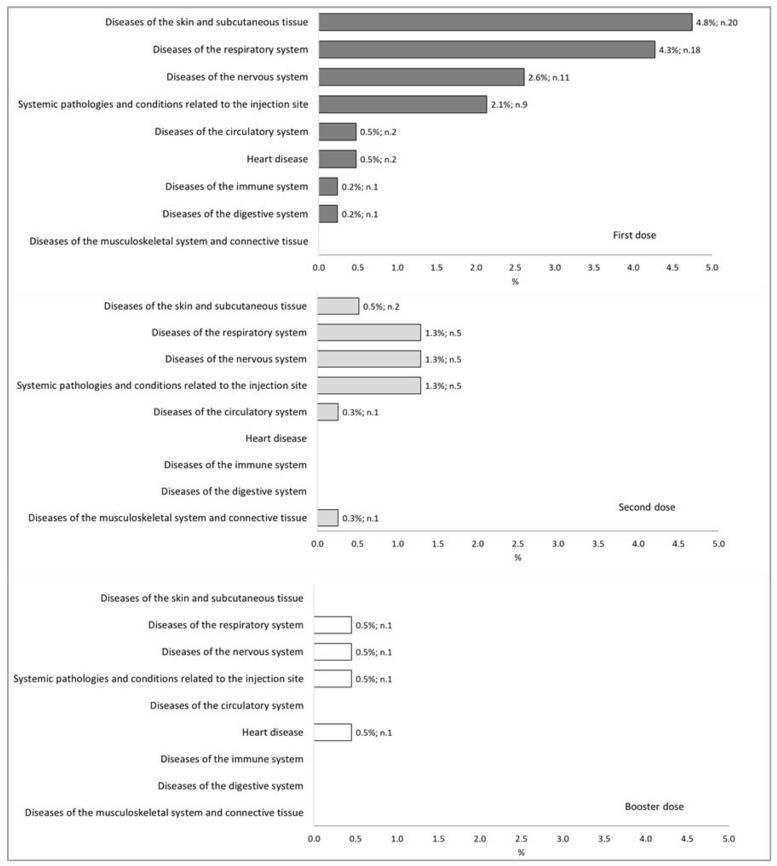
Distribution percentages of post-vaccination reactions classified through SOC classification, for each number of doses administered.

**Table 1 vaccines-11-00574-t001:** Distribution of number of reactions (SOC classification) vs. doses administered.

Number Reactions (SOC Classification)	First Dose		Second Dose		Booster Dose		Total	
	(n.421)		(n.388)		(n.241)		(n.1050)	
	*n*	(%)	*n*	(%)	*n*	(%)	*n*	(%)
0	383	(91.0)	376	(96.9)	238	(98.8)	997	(95.0)
1	20	(4.8)	6	(1.5)	2	(0.8)	28	(2.7)
2	11	(2.6)	5	(1.3)	1	(0.4)	17	(1.6)
3	6	(1.4)	1	(0.3)			7	(0.7)
4	1	(0.2)					1	(0.1)

**Table 2 vaccines-11-00574-t002:** Multivariate logistic analysis (y = at least one reported reaction).

Covariates		Doses Administered (n.1050)	Reactions		*p*	adjOR (CI95%)
			n.	(%)		
Sex						
	Males	293	11	(3.8)	ref	
	Females	757	42	(5.5)	0.553	1.20 (0.66–2.16)
Age (media ± DS)		53.0 ± 18.5	43.6 ± 14.9		<0.001	0.95 (0.94–0.97)
Doses						
	First	421	38	(9.0)	ref	
	Second	388	12	(3.1)	<0.001	0.25 (0.13–0.49)
	Booster	241	3	(1.2)	<0.001	0.12 (0.04–0.39)
Number of previous allergies (media ± SD)		2.3 ± 1.4	2.5 ± 1.2		0.987	1.00 (0.78–1.29)
Vaccine allergy						
	no	897	41	(4.6)	ref	
	yes	153	12	(7.8)	0.379	1.39 (0.67–2.89)
Food allergy						
	no	713	36	(5.0)	ref	
	yes	337	17	(5.0)	0.217	0.66 (0.34–1.28)
Medicine allergy						
	no	298	14	(4.7)	ref	
	yes	752	39	(5.2)	0.866	1.06 (0.52–2.19)
Other medicine with PEG allergy						
	no	681	34	(5.0)	ref	
	yes	369	19	(5.1)	0.856	1.07 (0.54–2.09)

## Data Availability

The data supporting the findings of this study are available from the corresponding author upon reasonable request, and first has to be approved by Public Health Department (ULSS 7, Veneto region).

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
