# Peer review of "Is SARS-CoV-2 Vaccination of Subjects with a Prior History of Allergies Dangerous? Experiences in the Veneto Region of Italy"

_vaccines, 2023, doi:10.3390/vaccines11030574_

Round 1
Reviewer 1 Report
The manuscript entitled “Sars-CoV-2 vaccination among subjects with a prior history of allergies is dangerous? An experience in the Veneto region” seems an interesting piece of work, however the following quarries have to clarify before publication.
Comments
1. Authors have mentioned that the subjects with a prior history of allergies have adverse effects on SARS-CoV-2 vaccination; if authors explain the types and nature of the prior history of allergies more in detail it could be more informative.
2. Is authors have analyse the co-morbidities of the patients those who are having adverse effects after vaccination
3. Is there any correlation with age and the severity of the adverse effects?
4. Authors requested to provide the standard observation period to study the adverse effect on the study population.
5. Are the patients who are included for this study all have the same types of vaccine at first, second and booster dose?
6. Authors requested to provide information on the nature of adverse effects on the patients whether all the patients have severe effect or mild to moderate effect after first dose of their vaccination.
7. Did authors have noticed any allergic symptoms in the patient's post-vaccine period?
8. The detailed abbreviations of some short forms are not provided in the manuscript and it makes reading the manuscript difficult.
9. I suggest authors, to replace the figures with good clarity or at least 600 DPI.
Author Response
- Authors have mentioned that the subjects with a prior history of allergies have adverse effects on SARS-CoV-2 vaccination; if authors explain the types and nature of the prior history of allergies more in detail it could be more informative.
Many thanks for the suggestion. I explain this better in lines 110–123.
- Is authors have analyse the co-morbidities of the patients those who are having adverse effects after vaccination.
Unfortunately, we didn’t analyze patients’ comorbidities for study purposes. This is cited as one of the limits of the study (line 297).
- Is there any correlation with age and the severity of the adverse effects?
Thanks for the observation. We added the sentence in the Results section, lines 211–213
4.Authors requested to provide the standard observation period to study the adverse effect on the study population.
The period is specified in the Materials and Methods section.
- Are the patients who are included for this study all have the same types of vaccine at first, second and booster dose?
Thanks for the advice. We added the information in the text, line 125.
- Authors requested to provide information on the nature of adverse effects on the patients whether all the patients have severe effect or mild to moderate effect after first dose of their vaccination.
In the Materials and Methods section, we clarified the criteria applied (line 137) and specified the severe events in Results (lines 206–210).
- Did authors have noticed any allergic symptoms in the patient's post-vaccine period?
Thanks for the suggestion. We added in this in the Results section (lines 179–187).
- The detailed abbreviations of some short forms are not provided in the manuscript and it makes reading the manuscript difficult.
Thanks. The corrections are inserted.
- I suggest authors, to replace the figures with good clarity or at least 600 DPI.
Done.
Reviewer 2 Report
The proposed hypotesis of this paper is very interesting and represents a very common question within all Units of Allergology. In fact, clinical trials on safety and tolerability of COVID-19 mRNA vaccines excluded patients with history of allergies and guidelines of management of these category of patients are often difficult to apply in the real-life. However, in my opinion, the weakness of this paper is the lake of objective data to establish if the adverse reactions to vaccines are really immune-mediated reactions. For example, the authors didn’t measure level of serum triptase of patients who experienced the reaction; moreover, it was not perform an allergological exam with the components/excipients of mRNA vaccines on patients with adverse reactions, not even before making a further dose of vaccine. In fact, the authors only considered the reaction latency to define it as allergic. The study also contemplates only a small reality, namely that of the Veneto Region.
This lack of objective evidences makes the findings and conclusions of this paper of uncertain scientific significance and soundness and it is required to be supported by further investigations in this field.
On the other hand, the authors already declare these potential weaknesses within the manuscript.
Author Response
Thanks for the comment. We agree, and so we added that our results required further investigation.
Round 2
Reviewer 1 Report
Revised version can be accepted for publication
Best wishes for the authors
Author Response
Many thanks. Best regards